# The Relationship Between Inflammation and the Development of Cerebral Ischaemia and Hypoxia in Traumatic Brain Injury—A Narrative Review

**DOI:** 10.3390/ijms26168066

**Published:** 2025-08-20

**Authors:** Alan Nimmo, Alexander Younsi

**Affiliations:** 1College of Medicine and Dentistry, James Cook University, Cairns, QLD 4870, Australia; 2Department of Neurosurgery, Heidelberg University Hospital, 69120 Heidelberg, Germany; alexander.younsi@med.uni-heidelberg.de; 3Medical Faculty, Heidelberg University, 69120 Heidelberg, Germany

**Keywords:** traumatic brain injury, inflammation, neuroinflammation, blood-brain barrier, cerebral oedema, intracranial hypertension, vascular stasis, capillary stalling, thrombo-inflammation, glymphatic system

## Abstract

Traumatic brain injuries (TBI) represent a leading cause of morbidity and mortality globally. Whilst clinical care has significantly improved in recent years, there is still significant scope to improve patient outcomes, particularly in relation to quality of life. However, there is a window of opportunity for clinical intervention, since most of the mortality and morbidity is associated with secondary injury processes that arise after the initial trauma. In the brain, as with any tissue, inflammation plays an important role in the response to injury. However, particularly with severe injuries, an excessive inflammatory response can have detrimental effects. Following TBI, inflammation can lead to the development of cerebral oedema and a rise in intracranial pressure. Without effective control, these processes can rapidly lead to patient deterioration. This narrative review focusses on the role of inflammation in TBI in order to examine the strategies that may help improve patient outcomes. Whilst there is clearly a relationship between the development of cerebral oedema, rising intracranial pressure (ICP), and poor patient prognosis, there are also discrepancies in terms of their impact on patient outcomes. In addition to causing a rise in ICP, this review examines in what other ways inflammation and the development of cerebral oedema may contribute to the injury process. The potential for these factors to impact upon microvascular function and reduce cerebral tissue perfusion and oxygenation is explored. In addition, the impact of TBI on glymphatic function is discussed. Following an evaluation of the potential injury processes, the scope for intervention and the development of novel therapeutic approaches is explored.

## 1. Introduction

Globally, traumatic brain injuries (TBI) represent one of the most significant causes of mortality and morbidity in the younger population, placing a huge burden on society, both socially and financially [1,2]. Whilst there have been significant improvements in both the prevention and treatment of TBI, there is still much work to be done, particularly in relation to improving quality of life outcomes [3,4]. TBIs represent a very heterogeneous set of injuries. However, it is recognized that there are a number of pathological processes that are common, not just to TBI, but also to other acute brain injuries, such as stroke [5,6]. With TBI, the only approach to managing the primary injury is prevention. Importantly, though, most of the ensuing mortality and morbidity is associated with the subsequent secondary injury processes [1]. As such, there is scope for clinical intervention to ameliorate the impact of these secondary injuries. One pathological process that is known to play a significant role in TBI is inflammation, and this can lead to a range of subsequent issues, both acute and chronic [7].

Inflammation is not only the most common pathological processes, but also one of the first to be documented, appearing in ancient Greek and Roman texts [6]. However, the recognition of the role of inflammation plays in a wide range of central nervous system (CNS) disorders is much more recent [7,8]. As in the periphery, inflammation can progress from being basically a protective mechanism to one that contributes to detrimental patient outcomes.

Whilst inflammation is associated with a diverse range of responses, altered microvascular function, including increased vascular permeability and leukocyte adhesion, is central to both the protective and detrimental aspects of inflammation [8]. Within the CNS, the inflammatory response to acute insults can lead to well-documented problems, such as cerebral oedema and raised intracranial pressure (ICP) [9]. However, as is seen in the periphery, inflammation may have broader effects on haemodynamics, which could also contribute to poor patient outcomes [10,11]. The aim of this narrative review is to examine the evidence as to how inflammation and the associated vascular responses may contribute to cerebral ischaemia and hypoxia following acute insult, such as a TBI.

## 2. Inflammation

The human body has developed a range of powerful defence mechanisms to deal with invasion by pathogenic organisms, and activation of these leads to the initiation of an acute inflammatory and immune response. As originally documented, the key signs of inflammation are that the affected area is hot, red, swollen, and painful [9]. As these signs suggest, the vascular system plays a key role in mediating an inflammatory response, as well as initiating both innate and adaptive immune responses [10,11]. The vascular responses, such as fluid exudation and leukocyte migration, align with the role of inflammation in response to infectious disease [12,13]. However, the exact same mechanisms are initiated in response to any noxious stimulus that leads to cell injury, including trauma [14,15]. All aspects of the inflammatory response, from initiation to progression and ultimately resolution, are orchestrated by a plethora of both pro- and anti-inflammatory mediators [16].

In general, acute inflammation primarily serves as a protective mechanism, providing an appropriate response to deal with an insult or infection, enabling tissue repair, and helping restore homeostasis [10,17]. As homeostasis is restored, the inflammation itself should resolve as part of that process [16,17]. Unfortunately, there are situations where inflammation changes from being a protective mechanism to becoming a disease process in its own right [14]. That transition of inflammation to a pathological condition is commonly seen when acute inflammation does not resolve and becomes chronic. Chronic inflammation is not just a common element of peripheral disease, but it underlies some of the most prevalent and significant neurological conditions, including multiple sclerosis, Alzheimer’s, and Parkinson’s disease [18].

Recognizing and addressing the role inflammation plays in pathological conditions can be a key aspect in improving their clinical management. This was clearly demonstrated by the improved management of asthma following its recognition as a chronic inflammatory condition in the 1980s [19,20]. However, the improved management of any inflammatory-related condition relies not only on understanding the role of inflammation in the disease, but also on identifying an appropriate approach to manage that inflammation [21]. This again can be illustrated by asthma, where corticosteroids can be lifesaving, whilst the other common anti-inflammatory agents, non-steroidal anti-inflammatory drugs (NSAIDs), may trigger an asthma attack in some individuals [22,23].

Whilst there is a significant focus on chronic inflammation as a pathological state, the potential for detrimental outcomes from acute inflammatory responses also needs to be considered. If there is either an excessive or an inappropriate activation of the acute inflammatory response, that too can have devastating consequences, as seen in situations like clinical shock and severe anaphylactic reactions [24,25]. One of key impacts of such an excessive inflammatory response is upon tissue perfusion and oxygenation [26].

## 3. Haemodynamic Changes with Inflammation

As discussed, inflammation is primarily a response associated with vascularized tissue and results in a range of haemodynamic changes. In addition to the localized vasodilation, there is an increase in vascular permeability, particularly associated with postcapillary venules, that leads to the extravasation of both fluid and serum proteins, including complement proteins and immunoglobulins [27,28,29]. The formation of that exudate, together with the influx of phagocytic cells to the affected area, represents an important element of the initial immune response, promoting the clearance of pathogens and cell debris [30]. Lymphatic vessels represent another important component of the vascular system’s role in inflammation. Lymphatic vessels provide a route for the drainage of that extravasated fluid, along with the removal of immune cells, cellular debris, and inflammatory mediators from the affected area [31,32]. The transport of this lymphatic fluid to the lymph nodes is central to the initiation of the adaptive immune response, whilst the removal of inflammatory elements from an affected tissue may also represent a step towards the resolution of inflammation [31].

The inflammation-induced vascular responses that lead to oedema formation impact upon normal capillary fluid dynamics. The movement of fluid across the walls of capillaries and postcapillary venules is regulated by both hydrostatic and oncotic pressures, commonly referred to as the Starling forces, whilst the capillary wall itself acts as a selectively permeable membrane [33]. Most capillary beds have a low permeability to plasma proteins, allowing for a slight leakage into tissue fluid, whilst normally the capillaries in the brain, with their tight junctions, are virtually impermeable to plasma proteins [12]. However, it is that permeability to serum proteins that significantly changes with inflammation, thereby disrupting the natural balance of the Starling forces, leading to a high interstitial oncotic pressure and the development of oedema [27]. The formation of oedema also leads to an increase in the hydrostatic pressure of the interstitial fluid [29].

Whilst the development of oedema with inflammation is a significant issue, the haemodynamic changes that lead to its formation can bring additional consequences. The increased fluid extravasation associated with inflammation leads to a local decrease of fluid in the vascular compartment. In turn, this leads to decreased blood volume, increased blood concentration and viscosity, decreased blood flow, and vascular stasis [34,35]. This local slowing of blood flow, together with the enhanced expression and activation of adhesion molecules, favours leukocyte extravasation from the post-capillary venules at the site of inflammation [13,36]. The microvascular narrowing associated with leukocyte adhesion in the post-capillary venules further reduces local blood flow [37]. It has been postulated that evolution has favoured venules as the site of leukocyte transmigration, since, if the same event occurred in capillaries, it could result in the complete occlusion of microvessels at the site of inflammation [28,37]. Finally, the local inflammatory response can lead to activation of the clotting cascade, fibrin deposition, and the formation of microthrombi [38]. Together, all these factors impact on microvascular perfusion in inflamed tissues, leading to hypoperfusion, ischaemia, and tissue hypoxia [39]. In clinical situations where there is a significant systemic inflammatory response, such as with burns and sepsis, the widespread impact on microcirculatory function may potentially lead to multiple organ dysfunction [35,40,41].

## 4. Inflammatory Responses in the Central Nervous System

As with peripheral tissues, the CNS may be impacted by both acute and chronic inflammatory responses [6,18]. Again, there is a need to achieve a balance between the beneficial aspects of inflammation (e.g., defence against pathogens and tissue healing) while limiting any detrimental aspects (e.g., tissue injury) [42,43,44]. With acute insults, such as TBI and stroke, an excessive acute inflammatory response is known to contribute to secondary injury processes such as cerebral oedema and raised intracranial pressure [45,46]. There is also growing evidence to suggest that excessive inflammation associated with these acute events can play an active role in the progression to neurodegenerative conditions [47,48,49].

As discussed, the vascular system plays a significant role in mediating inflammatory responses [28]. The brain, with its high metabolic activity and significant demand for oxygen, is one of the most highly perfused organs in the body, suggesting a potential susceptibility to an inflammatory reaction [50,51]. However, another factor to consider in relation to this is the blood–brain barrier (BBB). Historically, the CNS has been viewed as being immunoprivileged, being shielded from the peripheral immune system by the BBB [47]. In terms of immune protection, the resident microglial cells, through their expression of a range of pattern recognition receptors, are capable of detecting both pathogen-associated molecules and tissue damage [52,53,54,55]. Microglia, along with perivascular macrophages, also represent the phagocytic cells of the CNS [53,55,56]. Hence, there are resident cells that can provide immune protection within the CNS, although that response is, by nature, an acute innate response [52]. However, it is now known that glial-mediated responses represent only part of the picture in terms of CNS immune reactions. Instead of the BBB representing a physical barrier that separates peripheral and glial immune responses, it is now recognized as a dynamic, functional barrier that serves as an interface between the two [57]. As such, neuroinflammation is an inflammatory response within the CNS, orchestrated by both glial cells and peripheral immune cells [58].

The physical and functional integrity of the BBB is a critical factor in maintaining brain homeostasis, with part of that role being the regulation of immune and inflammatory responses. The BBB is composed of and regulated by complex interactions between a variety of cell types. The vascular endothelial cells (VECs), with their tight junctions, limit paracellular flow, whilst enabling selective transcellular movement via specific transporters and receptors [59,60]. Astrocytic end-feet, which are closely associated with the VECs, also express specific, polarized transport molecules that play a role in the regulation of solute and fluid movement [61]. The movement of water across the BBB, as well as the blood–cerebrospinal fluid barrier, is facilitated by the water channel, aquaporin-4 (AQP4). In terms of the BBB, AQP4 is highly localized to the astrocyte end-feet, where it is thought to enable the flux of water molecules between the brain parenchyma and blood by enhancing astrocytic transmembrane flux [62,63]. AQP4 is central to water homeostasis in the brain. On its own, AQP4 has the potential to allow for the bi-directional flow of water. However, the co-localization of AQP4 with other channels in the astrocytic membrane, such as the inwardly rectifying potassium channel, can allow for polarized function [63].

Astrocytes, through their end-foot processes, also provide a mechanism of communication between neurons and VECs [64,65]. The pericytes, embedded in the basement membrane of cerebral vessels, have extensive contacts with the VECs and astrocyte end-feet, as well as perivascular microglia, macrophages, and neurons [66,67]. Together, all these elements represent the neurovascular unit (NVU), a concept developed to encompass the anatomical and functional relationships between brain cells and the cerebrovasculature [60,68]. The NVU is considered to exert control over such functions as cerebral blood flow, BBB permeability, and immune responses [69]. Positioned at the interface between the systemic circulation and brain parenchyma, the NVU can monitor and respond to immune and inflammatory signals in both the systemic circulation and the CNS [66].

As in the periphery, the VECs of the cerebrovasculature mediate changes in vascular permeability, as well as the expression and function of adhesion molecules, allowing for leukocyte transmigration [70,71,72]. However, within the CNS, the regulation of these processes does appear to be more rigorously controlled. Pericytes also exert immunoregulatory functions, which may enable the integration of CNS and peripheral immune and inflammatory responses [66,67,73]. The numerous microglia within the environment of the NVU perform immunosurveillance and are capable of producing a wide range of pro-inflammatory mediators that may influence NVU function and BBB integrity [74].

In terms of inflammation in the CNS, there appears to be a two-way relationship between BBB dysfunction and neuroinflammation. Neuroinflammation and altered NVU function will impact BBB permeability and function, whilst altered BBB function may serve as a precursor for neuroinflammation [42]. BBB dysfunction is implicated in some of the most prevalent and significant neurological conditions, ranging from acute insults, such as TBI, through to chronic and neurodegenerative conditions, such as Alzheimer’s and Parkinson’s disease [75].

## 5. Glymphatic System

Whilst there are similarities between the periphery and CNS in terms of tissue fluid dynamics, there are also distinct differences. In the CNS, the extracellular, extravascular fluid comprises not just interstitial fluid (ISF) but also cerebrospinal fluid (CSF). The BBB, together with the blood–CSF barrier, normally exerts a rigorous control over the brain microenvironment to provide a stable environment for neuronal function [76]. The production of ISF is regulated by the BBB, whilst CSF is primarily formed by the choroid plexuses of the lateral ventricles [60,77,78]. Whilst these fluid compartments have physical and functional distinctions, there is also transport and exchange between the two spaces [79,80]. A conduit for that movement is the perivascular space surrounding blood vessels in the brain [81].

In both the periphery and CNS, ISF can return to the circulation via the venous system. In addition, in peripheral tissues, lymphatic vessels provide a mechanism for the removal of excess ISF, as well as waste products, including solutes and colloids. However, the parenchyma of the brain and spinal cord lacks an anatomically defined lymphatic vasculature. Instead, this function is fulfilled by what is known as the glymphatic system [82]. Whilst our understanding of the glymphatic system, both in health and disease, is still evolving, there is consensus around basic concepts [78,83]. There is a directional flow of CSF from the subarachnoid space to the brain parenchyma via the periarterial spaces. Subsequently, there is a movement of fluid between the perivascular and interstitial spaces that is facilitated by the AQP4 channels associated with astrocytic end-foot processes [84]. Within the parenchyma, there is a mixing of CSF and ISF, with this combined fluid draining out of the brain through the perivenous spaces. The fluid from the perivenous space can subsequently drain into the peripheral lymphatic system, passing through the superficial and deep cervical lymph nodes [80,85,86]. One unusual characteristic of the glymphatic system, as opposed to the lymphatic system, is that it is more active during sleep, including an anaesthetized state, a phenomenon that may have clinical implications [84,87].

CSF and ISF exchange, as well as glymphatic function, is considered to be important for brain homeostasis and health. An important aspect here is the glymphatic system’s role in clearing potentially toxic metabolites, such as β-amyloid and tau, from the brain [78,80,85]. In addition, it provides a direct route of communication between the CNS and peripheral immune system, allowing for immune surveillance of the CNS, thereby challenging the traditional view of its immune-privileged status [88,89,90,91]. Together, these findings have provided new insights into the pathological processes that may contribute to common neurodegenerative disorders [92,93].

## 6. Traumatic Brain Injury, Inflammation, and Cerebral Oedema

As indicated, TBIs represent a leading cause of death and disability, with much of that long-term or permanent damage attributable to the delayed, secondary injury mechanisms. Importantly, this provides a window of opportunity for therapeutic intervention [94,95]. As with any tissue injury, an acute inflammatory response is associated with TBIs, and again, that inflammatory response may represent a double-edged sword, either helping to resolve the issue or exacerbate it [96,97]. Whilst multiple factors contribute to the high mortality and morbidity associated with TBI, the development of cerebral oedema remains the most significant predictor of poor outcome [46,98,99]. As such, an understanding of the mechanisms associated with the formation of cerebral oedema, as well as the implications of the oedema itself, may help to guide the development and application of improved interventions.

The physical and functional integrity of the BBB is central to maintaining brain homeostasis. However, following acute insults, such as TBI, there can be a rapid loss of normal BBB function, and, in particular, the integrity of the endothelial tight junctions [71,98,100,101,102]. This altered BBB function may be both a component and a driver of the neuroinflammatory response [6,103]. Altered BBB function is also central to the development of vasogenic oedema, which arises due to inflammatory endothelial injury and the extravasation of plasma proteins and water into the interstitial space [24,46,102,104].

Along with vasogenic oedema, tissue ischaemia and hypoxia are common co-existing issues [97]. Not only are these factors on their own potentially detrimental, particularly to the brain, but they may also lead to a vicious cycle of events [97]. Tissue hypoxia has the potential to cause cellular energy failure, and with it, the loss of normal ion homeostasis. This can lead to increased intracellular Na^+^ ion concentrations, resulting in a shift of fluid into the intracellular compartment, and the development of cytotoxic oedema, characterised by cellular swelling due to ATP depletion and Na^+^/K^+^ pump dysfunction [105]. In turn, this can establish a gradient for Na^+^ and water to move from the vasculature into the brain’s extracellular space, even across an intact BBB, leading to ionic oedema. These mechanisms are sequentially and mechanistically distinct but often co-occur [46,105].

It is also proposed that CSF can contribute to ionic oedema, particularly in the anoxic brain, and that impaired CSF–interstitial fluid exchange may further exacerbate fluid accumulation [106]. In this context, the glymphatic system, which facilitates clearance of interstitial solutes via AQP4 channels on astrocytic end-feet, may also be disrupted—compromising fluid drainage and contributing to a state of CSF-shift oedema [78,107]. Table 1 provides a summary of the characteristics of the different forms of cerebral oedema.

Whilst there are distinctions between these different forms of cerebral oedema, they are also related, particularly through the inflammatory processes that drive them. Inflammation promotes BBB dysfunction (vasogenic), modulates ion channels like Sur1-Trpm4 (cytotoxic/ionic), and alters AQP4 expression and polarity, influencing water movement in both directions. Moreover, inflammation impairs glymphatic outflow, further hindering interstitial fluid clearance. Thus, inflammation, through its ability to promote and sustain oedema, directly affects tissue perfusion and oxygenation, which in turn worsens ischaemia and hypoxia [28,39].

In addition to oedema causing hypoxia, hypoxia can lead to inflammation and oedema formation. Tissue hypoxia has the potential to cause inflammation through the activation of the hypoxia-inducible transcription factors [110,111]. The impact of this pro-inflammatory pathway is perhaps most clearly seen with altitude sickness, where cerebral oedema represents one of the most significant issues [112].

There is a wealth of evidence to support the notion that following an acute brain injury, a vicious cycle involving neuroinflammation, oedema, and ischaemia/hypoxia may develop [97]. The development of such a pro-inflammatory cycle may not just impact upon the acute management of a patient, but, assuming the patient survives, it could also become a factor in the transition from acute to chronic neuroinflammation [113,114]. As such, there is value in trying to elucidate what factors may contribute to this relationship between oedema and ischaemia and to examine potential approaches to break that cycle.

## 7. Cerebral Oedema and Raised Intracranial Pressure

The development of cerebral oedema is the most significant predictor of poor patient outcome [46,98,99]. One of the key consequences of cerebral oedema is a rise in ICP. The Monro-Kellie hypothesis states that due to the rigid nature of the cranial vault, an increase in the volume of any one element within that vault, namely the brain itself, CSF, or the circulating blood, is going to lead to an increase in ICP [98,102]. The development of vasogenic oedema, through the extravasation of plasma and macromolecules into the interstitial space following BBB disruption, represents a significant potential driver of raised ICP [98].

However, not all forms of oedema contribute equally to ICP [113,115]. For instance, cytotoxic oedema, which involves intracellular water accumulation in response to ionic pump failure, results in a redistribution of water rather than a true increase in tissue volume [105]. In contrast, ionic oedema, which follows cytotoxic changes, involves water influx into the interstitial space and may contribute more significantly to mass effect [46]. Similarly, interstitial oedema caused by transependymal CSF flow and CSF-shift oedema linked to impaired glymphatic clearance can increase parenchymal volume, particularly in periventricular and subcortical regions, adding to ICP [106] (Table 1).

The impact of rising ICP, and the resulting intracranial hypertension (ICH), can inflict further damage to the brain and potentially become life-threatening. Given the rigid nature of the cranium, the pressures associated with ICH can impact severely on the delicate neural tissue [116]. The rising pressure within the cranium will also impact upon blood flow to the brain through a reduced cerebral perfusion pressure (CPP; CCP = mean arterial pressure—ICP), with a resulting reduction in brain oxygenation [98,117,118]. Ironically, as the brain’s autoregulatory mechanisms attempt to compensate for this reduced blood flow through vasodilation, this will further increase ICP and reduced CPP [119]. In addition, the physical swelling of brain regions can cause a shifting, or herniation, of the brain tissue, not just within cranial compartments but also between them [120,121]. For example, oedema and increased pressures from above can force the cerebellar tonsils into the foramen magnum, with a resulting compression of the medulla and the posterior inferior cerebral arteries [115]. Compression of the brain stem and, particularly, the respiratory centres produces additional life-threatening consequences [122].

Following the seminal work of Lundberg in 1960 [123], controlling ICP and maintaining CPP has been a major focus of the critical care of TBI patients. There have been strong rationales for this, given that raised ICP is a common factor in patients with severe TBI, and that a sustained rise in ICP is associated with poor patient prognosis in both the adult and paediatric patient population [124,125,126]. In terms of mortality, in approximately half of cases, severe intracranial hypertension is the primary cause of death [124,125,126].

There are two primary aims in terms of controlling ICP. The first is to maintain CPP, and the second is to prevent brain herniation. A range of approaches can be used to try and lower ICP and maintain CPP, including sedation, the use of hyperosmolar agents, and performing decompressive craniectomy [117,127]. These different approaches produce their effects through reducing the different factors that contribute to the raised ICP, such as cerebral blood volume, interstitial fluid volume, or removing the volume limitation of the cranial vault [117,127]. At face value, performing a surgical decompressive craniectomy should represent a relatively robust mechanism for reducing ICP, and whilst it is widely used, there are question marks around the actual benefits that are gained [117,128,129]. The critical care of TBI patients has certainly improved significantly in recent years, but most of the benefit has been delivered in terms of reduced mortality rates, rather than reducing morbidity [130]. The potential for decompressive craniectomy to reduce morbidity and produce tangible improvements in patient outcomes, rather than just reduce mortality, is still being questioned, as evidenced by Sahuquillo and Dennis’s systematic review [131]. As a result, there has been a movement towards using the measurement of brain tissue oxygen levels as a guide to patient management [132]. This approach evolved with the recognition that in some patients, there may be brain tissue ischaemia and hypoxia despite a clinical lowering of ICP, as well as maintenance of an effective CPP [133]. As with raised ICP, there is again a clearly recognized correlation between cerebral hypoxia and patient outcome in TBI [134,135,136]. Such observations do pose the question as to how these factors may be interrelated.

## 8. Intracranial Pressure, Oedema, and Cerebral Hypoxia

Whilst there appears to be two separate doctrines here, the weight of evidence supporting both suggests that there must be correlations and that both represent key elements in the secondary injury process. Understanding how ICP and brain hypoxia are correlated may help inform patient management and potentially help reduce the morbidity rates.

Both pre-clinical and clinical studies have shown that the development of cerebral oedema will lead to a rise in intracranial pressure, a decrease in cerebral perfusion pressure and brain tissue oxygenation [137,138]. However, there are also situations where there are discrepancies between these variables, suggesting that there may be a degree of independence between them. One explanation could relate to the underlying oedema subtype. For example, cytotoxic oedema may lead to microcirculatory impairment and hypoxia even before ICP rises significantly [46,105,106,115]. However, there is also evidence to suggest that the actual cause of the rise in ICP may also play a role in determining the outcome [124]. For example, when raised ICP results from a mass lesion, poor patient outcomes have been reported when the lesion causes a rise in ICP of >40 mmHg. However, with raised ICP associated with cerebral oedema, similarly poor outcomes have been reported when oedema results in an ICP elevation of only >10 mmHg [124]. In a similar vein, a study of patients suffering from idiopathic intracranial hypertension demonstrated that a rise in ICP in the order of 40 mmHg may not cause any permanent neurological deficit provided adequate CPP is maintained [139]. In contrast, cerebral ischaemia and low brain tissue oxygen levels may be seen with severe tuberculosis meningitis despite effective control of ICP and CPP [140].

As such, the “classical” view that raised ICP, as well as lowered CPP, is the primary driver of tissue hypoxia and ischaemia appears to be too simplistic a model. These findings support the idea that raised ICP alone is insufficient to explain regional hypoxia, especially in cases dominated by cytotoxic or ionic oedema, where impaired autoregulation and microvascular dysfunction—not bulk volume—are more prominent contributors [132,135]. Instead, these observations suggest that, in addition to ICP, there may be other factors associated with the formation of cerebral oedema that are also impacting upon brain oxygen levels. Patient data generated by Kirkpatrick and colleagues, using laser Doppler flowmetry (LDF), demonstrated that associations do exist between ICP, CCP, and cerebral microcirculatory flow. However, there were also instances where a dissociation occurred [138]. In one patient who did not respond to treatment, LDF showed a severe decline in cerebral microcirculatory flow prior to the rising ICP exceeding 40 mmHg. Preclinical data generated by Vink and colleagues, using a sheep model of TBI, also clearly demonstrated the inverse relationship between ICP and brain tissue oxygen levels [137]. However, the data also indicate that the significant decline in brain tissue oxygen levels appears to be more rapid than the steadily rising ICP, supporting the notion that in TBI, detrimental outcomes may be observed with relatively moderate rises in ICP [124].

It has long been recognized that traumatic brain injuries have a significant impact on cerebrovascular function. As suggested by Stocchetti and Longhi, alterations in microcirculatory function may be a critical aspect of TBI [136]. Normal control of cerebral blood flow is absolutely dependent upon autoregulation, with regional blood flow being directed to the more active areas in the brain through a process known as neurovascular coupling [141]. The neurovascular unit (NVU) plays a central role in this process [141,142]. Studies indicate that the normal autoregulatory response may be impaired in patients with a TBI, and that loss of autoregulatory function is another key prognostic indicator [143,144]. In addition, the NVU also plays a critical role in the acute neuroinflammatory response and the changes in BBB function. The loss of BBB integrity with a TBI is a significant component of the neuroinflammatory response [145,146]. The impact of this on microvascular function may be similar to what is seen in the periphery [25,35,71]. It would appear that the function, or dysfunction, of the NVU may be crucial in the acute response to TBI, both in terms of the control of cerebral blood flow and the integrity of the BBB [6,147,148].

As in the periphery, the acute inflammatory response will have a significant impact on the cerebral microvasculature, with the vasculature then subsequently, and actively, participating in the secondary injury process [35,149]. Increased BBB permeability and fluid extravasation will favour the development of vascular stasis, and this, together with the expression and activation of adhesion molecules, will facilitate leukocyte recruitment [8,39,150]. In the CNS, the phenomenon of capillary stalling has recently been described, particularly in relation to the pathological mechanisms underlying neurodegenerative disorders, such as Alzheimer’s disease [151,152,153]. Whilst the local regulation of blood flow has been considered to be the domain of arterial vessels, recent studies have suggested that in the brain, capillaries may also play a role in autoregulation. Unlike arterioles, where vessel diameter is controlled by vascular smooth muscle, the flow of blood, and, particularly, its cellular components, through these capillaries is influenced by the surrounding pericytes [50,154]. This capillary stalling may happen as a result of leukocytes occluding the microvessels, resulting in decreased local cerebral blood flow [151]. These changes in cerebral blood flow, including capillary stalling, have been observed in an animal model of systemic inflammation [155]. The innate inflammatory response also favours the activation of the coagulation system, which can lead to thromboinflammation [149,156]. This, together with local tissue hypoperfusion, can lead to platelet activation and adhesion, and the formation of microthrombi in the cerebral microvasculature [157,158,159]. In turn, this will lead to regional ischaemia, amplifying the local inflammatory response mediated by the cerebral endothelial cells and the NVU [150].

Another factor that may compound the problem is interstitial oedema. Studies looking at the oxygen extraction fraction in patients with closed head injuries have demonstrated that the increased diffusion barriers may reduce cellular oxygen delivery and limit the capacity of the brain to increase oxygen extraction in response to hypoperfusion [135]. This diffusion barrier is especially prominent in interstitial and CSF-shift oedema, where periventricular white matter expansion may blunt oxygen delivery even when macrovascular perfusion appears preserved [106,109,132]. Hence, the inflammatory mechanisms associated with the development of cerebral oedema have the potential, in themselves, to cause regional ischaemia and hypoxia before allowing for any potential involvement of raised ICP.

In terms of cerebral oedema, it is also worth considering whether the glymphatic system may have a potential role in TBI. There is evidence for a relationship between inflammation and impaired glymphatic activity, particularly in relation to neurodegenerative disorders [78,90,160]. A number of studies have also highlighted a potential role for the glymphatic system in pathology of TBI [161,162,163]. In the periphery, enhanced lymphatic drainage helps resolve inflammation-associated oedema [31]. Increased interstitial fluid pressure has been shown to lead to the dilation of initial lymphatic vessels, thus facilitating drainage [32]. In contrast, in the CNS, there is evidence for impaired fluid efflux through the glymphatic system, especially in CSF-shift oedema, which may exacerbate parenchymal swelling and limit resolution [46,160,161]. Another factor here is that, naturally, glymphatic flow is not consistent. The impact of trauma and the excessive release of noradrenaline will lead to a suppression of glymphatic flow [161,164,165]. As well as contributing to oedema, the other impact of reduced glymphatic flow is a reduced clearance of proteins, such as amyloid-β and tau, potentially predisposing the injured brain to chronic disease [166]. Hence, there is a range of factors that can contribute to the development and severity of ischaemia and hypoxia associated with vasogenic cerebral oedema, as summarized in Figure 1.

When it comes to considering patient prognosis following a moderate-to-serious TBI, there is a need to try and develop a unifying theory that attempts to encompass the range of factors that are considered to contribute to the secondary injury process. Firstly, the involvement of raised ICP needs to be recognized, not only for its potential to lower CPP and cause brain herniation, but also its capacity to drive the hypoxia/ischaemia/swelling cycle that exacerbates the secondary injury process [97,137,138]. However, the anomalies associated with raised ICP also need to be considered [117,128,129,130,167]. This is particularly important in relation to TBI, since the cycle of deterioration can begin with relatively moderate rises in ICP [124]. If the only impact of post-TBI oedema was upon ICP, then the problem should be completely relieved by craniectomy, which unfortunately does not appear to be the case [128,131].

It appears that the cause of the rise in ICP is a significant determining factor [124], with the development of cerebral oedema being a key element [46]. This suggests that other factors associated with the development of oedema could also be contributing to patient outcome. One potential factor here could be the impact that oedema formation has on microvascular function. Fluid extravasation, due to the altered function of the BBB, will favour stasis in the microvasculature, facilitating leukocyte adhesion and extravasation [28,39,168]. The altered blood flow in the microvessels will also reduce oxygen and nutrient homeostasis in the affected area [138,150,151,168]. In addition, the stasis, combined with the inflammatory activation of the coagulation cascade, will favour the formation of microthrombi in the cerebral vasculature, which will also contribute to the regional ischaemia and hypoxia [113,157]. That hypoxia could also be exacerbated by the expansion of the interstitial fluid, impacting upon oxygen diffusion [135]. Compounding these issues, there is a loss of normal autoregulatory control over regional cerebral blood flow [141,169,170]. All these events, particularly the tissue hypoxia, will start to drive a pro-inflammatory cycle that will exacerbate the neuroinflammation and oedema [110]. The cerebral endothelial cells of the BBB appear to play a central role in initiating and regulating inflammation in response to ischaemia [150]. Finally, whilst increased glymphatic drainage could provide a mechanism to help ameliorate cerebral oedema, there is evidence that fluid efflux is reduced in TBI [161,165]. Hence, whilst there is almost certainly a cyclical relationship between oedema, raised ICP, and ischaemia/hypoxia (Figure 2A) [97], there are probably a range of interrelated factors that also contribute to patient outcome (Figure 2B).

As outlined in Figure 2B, in addition to that cycle involving a rise in ICP that is associated with cerebral oedema, the mechanisms associated with the development of cerebral oedema have additional implications that can also drive that cycle of patient deterioration. The impact of TBI, not just on the cerebral circulation, but on the microcirculation, needs to be considered, particularly in relation to brain tissue oxygen concentration [136,149,168]. The stasis associated with inflammation may not just impact directly upon tissue perfusion but may also lead to the formation of microthrombi [157,168]. In addition, the impact of cerebral oedema on oxygen diffusion will further exacerbate the tissue hypoxia [135,136]. Finally, the impact of injury on glymphatic drainage may represent an additional factor in relation to oedema formation and resolution [161,165]. The relationship between the secondary injury mechanisms proposed in Figure 2B is not intended to detract from the importance of patient intervention being guided by ICP and brain tissue O_2_ monitoring, but rather to be thought-provoking in relation to evaluating treatment strategies.

## 9. Potential Strategies to Target Cerebral Oedema and Its Implications

Historically, managing raised ICP and the associated reduction in CPP has been a primary target for the clinical management of TBI victims, since this would seem to provide the best guide for improving patient outcome. However, whilst surgical decompressive craniectomy has been shown to reduce the mortality rates associated with severe TBIs, unfortunately, there is not a similar improvement in relation to morbidity [128,131]. One interpretation of this is, that whilst surgery will manage the raised ICP and lowered CCP (Figure 2A) associated with oedema, there are still multiple repercussions that remain, and these may impact on brain tissue perfusion. Hence, the morbidity that persists following decompression could be associated with the other injury processes linked to oedema formation (Figure 2B).

A full evaluation of all the potential secondary injury processes associated with TBI is beyond the scope of this review, as is an evaluation of all the potential therapeutic strategies that have been, or are being, trialed. There are a number of excellent reviews and summaries of these aspects [1,3,4,45,171,172]. Instead, we will primarily review the evidence that may help underpin the proposed injury model outlined in Figure 2B and consider how the inflammatory response to injury may contribute to brain tissue hypoxia.

### 9.1. Inflammation

As discussed, inflammation is a common tissue response to injury, and the brain is no exception [147]. Unfortunately, despite the fact that inflammation is the most common pathological process, the range of clinically available anti-inflammatory agents is rather limited. In addition, there is not a standard balance between the efficacy and side effects of these agents [6,164,172]. Variations in the nature of the inflammatory response mean that sometimes an agent may be beneficial, and other times detrimental [22,23]. The same is certainly true in the brain. Whilst the use of corticosteroids was originally considered to be beneficial in TBI, the large-scale Corticosteroid Randomization after Significant Head Injury (CRASH) trial demonstrated that there was a significant increase in mortality in the methylprednisolone-treated group compared to the placebo control group [172,173]. There are also reported concerns regarding the use of NSAIDs, primarily as analgesics, in TBI. One of the concerns relates to the renal side effects of NSAIDs, and the potential to cause hyponatraemia or acute kidney injury. Whilst the significance of these effects remains to be determined [174,175], other than an analgesic effect, there is little evidence for any potential long-term benefit [176]. Indeed, studies looking at capillary stalling in models of Alzheimer’s disease indicate that prostaglandin E_2_ may be a mediator of capillary dilation, raising the possibility that NSAIDs could increase capillary stalling [153].

A number of other agents with putative anti-inflammatory activity, such as progesterone and cyclosporine, have been trialed [177]. Whilst Phase 2 studies with progesterone seemed to promise benefits, unfortunately, these were not borne out by the larger Phase 3 ProTECT III study [178].

Whilst few would dispute the need to develop effective anti-inflammatory agents for TBI and other neurological conditions, there are complicating factors [3,4,179]. In terms of oedema, there are multiple mechanisms by which inflammation can promote oedema, including endothelial dysfunction and BBB breakdown (vasogenic oedema), modulating ion channel expression such as Sur1-Trpm4 (ionic oedema) and altering astrocytic water transport via AQP4 (cytotoxic oedema). Therefore, anti-inflammatory interventions may have differential effects depending on the oedema context and timing. At the scientific level, there is a need to better understand neuroinflammation and the role of the NVU [6,58,180,181]. On one hand, this requires us to develop a better understanding of inflammation in the CNS, but it may also provide scope for novel developments [58,181].

### 9.2. Cerebral Oedema

Cerebral oedema is another factor that appears to be central to the injury process in TBI and is a primary contributor to raised ICP [117]. Unfortunately, to date, there is no optimal intervention for the effective management of cerebral oedema [98]. As outlined in Table 1, cerebral oedema may develop through a number of different mechanisms, and all of these may play a role in the oedema associated with TBI. These subtypes differ in their timing, compartment involvement, and relevance to ICP and tissue oxygenation. However, altered function of the BBB, and its role in the development of vasogenic oedema, is of interest, not just in terms of the fluid shift, but how it may enable the movement of plasma proteins and leukocytes into the CNS, and its potential to cause microvascular stasis [168,182].

One approach to try and reduce cerebral oedema is the infusion of hyperosmolar agents, such as hypertonic saline or mannitol. Whilst clinical studies have shown a reduction in ICP with hyperosmolar agents, unfortunately, there is no clear indication that their use reduces mortality or improves functional outcome [98]. However, the osmotic action of a hyperosmolar agent, such as sodium chloride (NaCl), is reliant on normal function of the BBB, and for the solute to have a low BBB permeability. With an intact BBB, NaCl will provide a driving force for the diffusion of water from the interstitium to the vasculature. However, this will not occur in areas where BBB integrity is altered or compromised. As such, it will be ineffective in regions of the brain impacted by vasogenic oedema. This underscores the importance of oedema subtype in determining therapeutic response: hyperosmolar agents are primarily effective in ionic oedema, but much less so in vasogenic or hemorrhagic contexts where the osmotic gradient collapses. Hence, hyperosmolar agents will primarily produce their osmotic action in non-affected areas of the brain, where normal BBB function remains [183]. This action will lower ICP, in line with the Monro-Kellie hypothesis, but it will not reduce the formation of oedema in the regions of the brain impacted by the TBI, where BBB function is compromised [98,183].

Another approach taken to reduce oedema and lower ICP is sedation. Sedation is normally achieved using either the benzodiazepine, midazolam, the intravenous general anaesthetic, propofol, or an opioid, such as fentanyl [98]. The rationale behind sedation is that reduced brain oxygen requirements may reduce blood flow and, hence, blood volume in the cranium, thereby reducing ICP. In addition, effective analgesia and sedation may protect against systemic factors, such as hypertension, that could potentially raise ICP [127]. The use of sedation is recommended for the management of moderate to severe TBI patients to help achieve ICP control [127]. Whilst sedation is probably superior to osmotic therapy, there is still much scope to improve patient outcomes [98].

In terms of managing BBB dysfunction and vasogenic oedema in TBI, a more targeted approach may be required [184]. Given AQP4’s central role in brain water homeostasis, it represents one potential target. It was postulated that drugs that could inhibit the function of AQP4, such as AER-271, would potentially reduce the development of cerebral oedema [98]. Unfortunately, the potential for AQP4 to allow for bidirectional water flow appears to complicate the situation. Studies using AQP4 knockout mice, or AQP4 inhibition, have shown that with some injuries, the loss of AQP4 function can lead to the reduced development of cerebral oedema. However, in other situations, this may actually exacerbate the oedema, suggesting that with vasogenic oedema, AQP4 may help clear fluid from the extracellular space [62]. In light of this, developing a capacity to manipulate the polarity of AQP4 could represent a better strategy [62,63].

The repurposing of an existing therapeutic agent has been explored as an approach to manage cerebral oedema. These include the oral hypoglycaemic agent, glibenclamide, the loop diuretic, bumetanide, and the vasopressin antagonist, conivaptan. Glibenclamide acts on a cation channel (SUR1-TRPM4) that is expressed in the CNS in a number of neuropathologies, including TBI [185]. SUR1-TRPM4 allows for an inward flow of Na^+^ ions, which in turn drives a flow of water through AQP4 channels. Acting on the sulfonylurea binding site (SUR1), glibenclamide has an inhibitory action on Na^+^ ion and water influx, thereby decreasing oedema formation. This mechanism is particularly relevant to cytotoxic and ionic oedema, where inflammation-induced upregulation of SUR1-TRPM4 contributes to endothelial swelling, BBB stress, and tissue hypoxia. Following on from positive preclinical data, there were also promising data generated from a few small clinical trials [186]. A large Phase 2 trial of an intravenous formulation of glibenclamide (BIIB093) was initiated, but, unfortunately, it was terminated on the basis of strategic considerations (https://clinicaltrials.gov/ct2/show/NCT03954041, accessed on 10 August 2025).

Like glibenclamide, the loop diuretic bumetanide acts on an ion cotransport protein, in this case NKCC1, which can also facilitate water diffusion through AQP4 [187]. NKCC1 has a natural role in regulating cell volume, but in the case of cellular energy failure, it can lead to an inward ion and water flux that results in cytotoxic oedema. Whilst some preclinical studies have shown positive results, there are limitations on the entry of bumetanide into the CNS due to efflux transporters, limiting its potential value [188,189].

A physiological regulator of water flux via AQP4 is antidiuretic hormone, or vasopressin. The V1a vasopressin receptor is constitutively expressed in the CNS, but is upregulated following TBI, where it may play a role in cerebral oedema [98]. Selective vasopressin V1a receptor antagonists, such as SR49059 or AVN576, have been shown to reduce cerebral oedema in animal models [190,191]. A small clinical study using the non-selective vasopressin antagonist, conivaptan, demonstrated the potential for such an agent to lower ICP, but, unfortunately, that action was accompanied by a significant diuretic effect [192].

Another natural mediator that is associated with the development of oedema is the neuropeptide, substance P (SP). Acting via its NK1 receptor, SP has a range of biological activities, including a pro-inflammatory effect. In terms of inflammation, stimulation of the NK1 receptor can lead to activation of the transcription factor, nuclear factor-κB (NFκB), leading to the production of key pro-inflammatory cytokines [193]. SP, again acting via the NK1 receptor, also has direct effects on the vasculature, particularly post-capillary venules, where it increases vascular permeability and promotes leukocyte adhesion and transmigration [194,195,196]. Both substance P and its NK1 receptor are abundant and widely distributed in the brain [197]. Following a demonstration that a neurogenic inflammatory response was associated with the development of cerebral oedema and neurological deficits following a TBI, these effects were subsequently shown to be associated with the release of SP and its action on the NK1 receptor [148,198]. Importantly, these effects could be attenuated through the post-injury administration of an NK1 receptor antagonist [182,198,199,200]. Using a large animal model of TBI (sheep), post-injury administration of an NK1 antagonist led to a decrease in ICP and a restoration of brain tissue oxygen levels [102]. Similarly, in a stroke model, the post-stroke administration of an NK1 antagonist gave a comparable lowering of ICP as compared to performing decompressive craniectomy [201]. Clinical studies have shown that following TBI, there is a significant rise in serum SP levels, and with severe TBI, there is an association between serum SP levels and mortality [104]. Together, these results led to the clinical development of a novel NK1 antagonist that was suitable for intravenous administration, EUC-001 [202]. EU-C-001 is currently in a multinational Phase 2 clinical trial for the treatment of moderate to severe TBI (www.clinicaltrialsregister.eu/ctr-search/trial/2017-004890-15/GB, accessed on 10 August 2025).

### 9.3. Microvascular Stasis

In the periphery, microvascular stasis is a recognized phenomenon with inflammation and oedema formation, and the preclinical evidence indicates that it is a similar picture in the CNS [35,150,155]. The reduced microvascular perfusion directly impacts on tissue oxygen delivery [35,153]. This perfusion deficit can be further compounded by oedema-related effects: cytotoxic and ionic oedema compress capillaries from within, while vasogenic and interstitial oedema raise interstitial pressure, narrowing capillary lumens externally. These mechanisms reinforce the link between oedema and stagnant microcirculation. Whilst strategies such as hyperosmolar therapy should enhance microcirculatory flow, again, at the site of inflammation, this will be affected by BBB integrity [203]. Bragin and colleagues have demonstrated that drug-reducing polymers (DRPs) can enhance microvascular perfusion in an animal model of TBI [168]. They showed that whilst injury resulted in decreased microcirculatory flow and associated tissue hypoxia, the post-injury administration of DRPs mitigated capillary stasis, leading to improved microcirculatory flow and tissue oxygenation.

### 9.4. Microthrombus Formation and Bleeding

Following a TBI, there appears to be a complex co-existence of both hypocoagulable and hypercoagulable states that can result in secondary injury via both haemorrhagic and ischemic lesions [158]. Both microthrombus formation and bleeding are detrimental to patient outcome, and their polar nature complicates patient management [204].

There is a recognized link between inflammation and thrombus formation, and this can be associated with BBB dysfunction [156,157,205,206]. The development of microthrombi following TBI will lead to impaired microcirculatory function and brain tissue oxygenation [204]. This may create a feedback loop, where microthrombosis worsens tissue hypoxia, leading to ionic imbalance and cytotoxic oedema, which in turn further compromises perfusion and promotes a pro-thrombotic endothelial phenotype. Animal studies have shown that the early administration of heparin has the potential to reduce leukocyte-mediated cerebral oedema after TBI [207]. The use of heparin brings the concerns of a haemorrhagic risk, but these may be mitigated by the use of low-anticoagulant heparin. A study using O-desulfated heparin demonstrated its potential to improve functional outcome without an increased risk of bleeding [208].

Tissue plasminogen activators (tPAs), with their thrombolytic activity, are used in the treatment of ischaemic strokes to enable tissue reperfusion. Unfortunately, they can also increase the risk of haemorrhage. In addition to their thrombolytic activity, tPAs may facilitate neuronal plasticity [209]. In an animal model of TBI, the intranasal administration of tPA has been shown to improve both motor and cognitive functional outcomes following injury. The benefit of the tPA in this situation could potentially be due to its neuroplasticity activity, its thrombolytic activity, or both [210].

In contrast, tranexamic acid is an antifibrinolytic agent that can be used to reduce bleeding. In the trauma setting, the early administration of tranexamic acid has been associated with reduced mortality in patients considered at risk of bleeding [1]. These findings led to the large CRASH-3 trial. This substantial trial demonstrated that the early administration of tranexamic acid reduced head injury-related deaths in patients with mild and moderate head injuries, but not in those with severe injuries [211]. The trial also showed that there was no increase in disability in survivors treated with tranexamic acid.

### 9.5. Glymphatic Drainage

Multiple studies have highlighted a potential role for the glymphatic system in the pathology of TBI [161,162,163]. This is particularly relevant in CSF-shift and interstitial oedema, where impaired perivascular clearance leads to fluid stagnation in the parenchyma. Glymphatic failure may also amplify the secondary injury by delaying the removal of extracellular fluid, metabolic waste, and inflammatory mediators. Importantly, reduced glymphatic flow will also reduce the clearance of toxic proteins, such as amyloid-β and tau [166].

In terms of TBI, the traumatic event itself leads to the excessive release of noradrenaline, which in turn leads to a suppression of glymphatic flow and lymphatic drainage [161,164]. Using a mouse model of severe TBI, Hussain and colleagues demonstrated that administering a post-injury cocktail of both alpha- and beta-adrenoreceptor antagonists led to the amelioration of cerebral oedema through improved glymphatic function. This, in turn, led to improved functional outcomes, both motor and cognitive [161].

In the periphery, lymphatic vessels undergo pronounced enlargement in response to inflammation. Vascular endothelial growth factor C (VEGF-C) appears to play an important role in the lymphatic response to inflammation. Blocking VEGF-C signaling exacerbates inflammation, whilst enhanced VEGF-C activity can reduce chronic inflammation [31]. Using a murine model of stroke, Boisserand and colleagues have demonstrated that the pre-injury administration of VEGF-C had a prophylactic action, reducing neuroinflammation and improving fluid drainage [89]. Whilst VEGF-C treatment may not readily translate into a post-injury therapy, it does highlight the potential value of improving glymphatic function.

Finally, it is recognized that, as with sleep, glymphatic flow may be increased during anaesthesia [212]. As such, one could speculate that an additional benefit associated with the use of sedation in TBI patients could relate to improved glymphatic drainage. In saying this, the choice of sedative or anaesthetic likely becomes important, since some agents, such as ketamine will increase flow, whilst others, such as isoflurane, decrease it [212].

## 10. Conclusions

Despite extensive preclinical and clinical research, TBI still represents a leading cause of morbidity and mortality globally [1,4,213]. Since much of the mortality and morbidity associated with TBI is due to the secondary injury processes that arise after the initial trauma, there remains scope for improving therapeutic interventions [1,98]. Inflammation is recognized as a key secondary injury process in TBI. In the CNS, one important impact of neuroinflammation is a loss of BBB integrity and subsequent development of vasogenic cerebral oedema. Cerebral oedema may lead to a rise in ICP and the development of intracranial hypertension. Rising ICP will start to impact on CPP, which can reduce cerebral blood flow and cause tissue herniation. This in itself will give rise to an injury cycle that can lead to patient deterioration (Figure 2A) [97,117]. Rising ICP in TBI patients represents a key prognostic indicator [117].

However, whilst intracranial hypertension is certainly a key factor in determining patient outcome, there are anomalies here, too. Surgical decompressive craniectomy can be effective in lowering ICP and helping to maintain CPP. However, whilst this approach may reduce the mortality rate associated with moderate to severe TBIs, it does not produce the same level of benefit in terms of morbidity. As such, it is important to explore what other injury processes may be contributing to patient outcomes. Other key prognostic indicators include the development of cerebral oedema itself, declining brain tissue oxygen levels, and loss of the autoregulatory control of cerebral blood flow. In Figure 2B, we have tried to develop a model to explain how, in addition to the raised ICP cycle, the development of cerebral oedema may impact on cerebral perfusion, particularly at the microvascular level.

From our understanding of both the periphery and CNS, the development of oedema may lead to vascular stasis and leukocyte extravasation, particularly in post-capillary venules. This, combined with the inflammatory reaction, can contribute to the formation of thrombi in the microvasculature. Whilst not yet reported in TBI, cerebrovascular capillary stalling is a recently recognized phenomenon, which may also be associated with inflammation [153]. All these factors may contribute to reduced tissue perfusion and, hence, reduced brain tissue oxygen levels. The expansion of ISF associated with the formation of oedema may also impact upon the rate of oxygen diffusion in the brain, further compounding tissue hypoxia. Whilst the glymphatic system should provide a mechanism to help resolve the accumulation of tissue fluid, unfortunately, glymphatic drainage may also be compromised in TBI.

In terms of therapeutic approaches aimed at improving patient outcomes following TBI, the schema proposed in Figure 2B may indicate some potential approaches. Novel anti-inflammatory agents that are effective in controlling neuroinflammation would represent the best option, not just for TBI, but for a range of neurological conditions. Unfortunately, such agents are probably far off. The next “common” process is the development of cerebral oedema. The BBB has long been recognized as an important therapeutic target, and there are a number of agents being developed that may help ameliorate cerebral oedema at this level. These include intravenous glibenclamide (BIIB093) and the intravenous NK1 antagonist, EU-C-001 [202,214]. Reduced oedema formation should improve microvacular perfusion, improve oxygen diffusion, and reduce the risk of microthrombi. However, in terms of preventing microthrombus formation, the use of low-coagulation heparin may represent an additional strategy. Finally, improving glymphatic draining may not just help relieve cerebral oedema but also prevent the accumulation of toxic proteins, such as amyloid-β and tau, and potentially reduce the risk of chronic inflammation and neurodegeneration [166]. There is preclinical evidence that this may be achieved through administering a cocktail of existing adrenoreceptor antagonists.

## 11. Future Directions

As the field advances, a more mechanism-based approach may help optimise patient outcomes in TBI. Recognizing the oedema subtype (cytotoxic, ionic, vasogenic, interstitial, and CSF-shift) may allow therapies to be tailored to the dominant mechanism and injury phase. Sur1-Trpm4 inhibitors, NK1 antagonists, anti-inflammatory agents, or glymphatic-targeted strategies could thus be selectively deployed. Ultimately, this might shift treatment paradigms away from ICP reduction alone toward mechanism-informed, personalised interventions targeting oedema biology itself.

There is still significant scope for further preclinical and clinical research in this area. From the issues discussed here, an improved understanding of the NVU and the cerebral microcirculation may provide new avenues for clinical intervention. Whilst the phenomenon of capillary stalling has been reported in relation to neurodegeneration, there would be value in seeing whether it is also associated with TBI. Given that prostaglandin E2 appears to be involved in capillary dilation, this may not only open up a new therapeutic window but also suggest that some anti-inflammatory agents could be contraindicated in TBI. The potential for drug therapies to improve glymphatic drainage also needs to be further explored. From preclinical research, there is evidence that existing therapeutic agents (adrenergic antagonists) could be utilized to improve glymphatic function. Finally, given the significance of vasogenic oedema in TBI, evaluating novel agents that may maintain or restore BBB integrity remains a key target. In terms of current translational research, the clinical development of NK1 receptor antagonists may deliver benefits in this area.

## Figures and Tables

**Figure 1 ijms-26-08066-f001:**
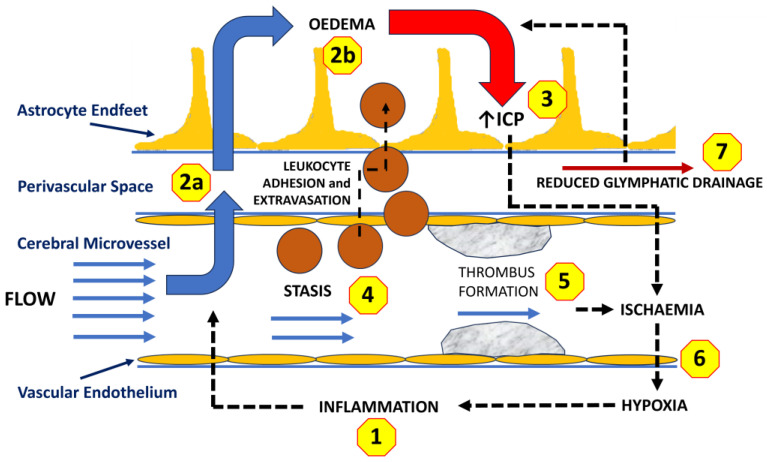
Schematic representation of how the development of oedema may impact upon the cerebral microcirculation, leading to brain tissue ischaemia and hypoxia. A cerebral microvessel with its associated perivascular space is represented. (**1**) An acute inflammatory reaction is initiated in response to an injury. (**2a**) Inflammation results in an increase in vascular permeability and the development of vasogenic oedema (**2b**). (**3**) Increased interstitial fluid volume will lead to a rise in intracranial pressure. (**4**) The decrease in intravascular fluid volume and flow, together with the adhesion of leukocytes to the vascular endothelium, will lead to vascular stasis. (**5**) The combined effects of inflammation, stasis, and increased vascular adhesion will increase the risk of microthrombus formation. (**6**) The reduced microvascular flow will result in ischaemia and hypoxia in surrounding tissues. That hypoxia will, in turn, exacerbate the inflammatory response (**1**). (**7**) Reduced glymphatic drainage will, in turn, increase interstitial fluid volume, exacerbating both the rising intracranial pressure and tissue hypoxia.

**Figure 2 ijms-26-08066-f002:**
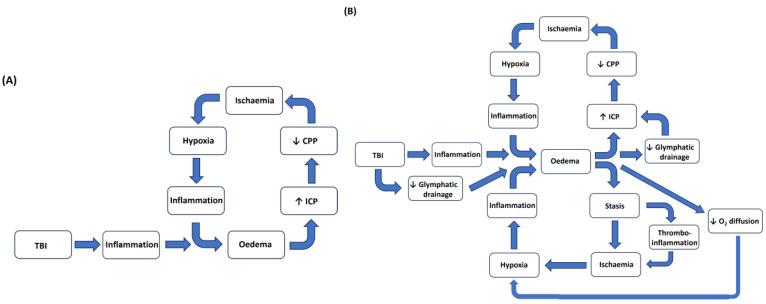
Panel (**A**) represents the generally accepted view of how cerebral oedema may contribute to a cyclical deterioration of brain perfusion and tissue oxygenation. Panel (**B**) builds upon this concept, whilst introducing the other factors that may also compound these issues, such as vascular stasis, reduced glymphatic drainage, reduced oxygen diffusion, and the development of microthrombi.

**Table 1 ijms-26-08066-t001:** A summary of the main subtypes of cerebral oedema and their characteristics.

Main Categories of Cerebral Oedema			
Category	Mechanism	Contribution to ICP	References
Vasogenic	Increased interstitial fluid volume associated with increased BBB permeability and the extravasation of plasma proteins and water into the interstitial space	Potential to cause a significant increase in ICP due to increased interstitial fluid volume	[24,46,102,104]
Cytotoxic (cellular)	Increased intracellular fluid volume—primarily associated with changed osmotic gradients associated with cellular energy failure and Na^+^/K^+^-ATPase dysfunction	Little direct impact on ICP, since it is associated with a fluid shift from the extracellular to intracellular compartment	[46,98]
Ionic (osmotic)	The osmotic forces that drive cytotoxic oedema, in turn, drive the movement of extracellular water and ions into the interstitial space—this may come from the vasculature or CSF	Potential to cause a significant increase in ICP due to increased interstitial fluid volume	[46,108]
Interstitial (CSF-shift)	Associated with transependymal CSF flow and CSF-shift oedema linked to impaired glymphatic clearance	Increased ICP associated with increased parenchymal volume, particularly in periventricular and subcortical regions	[106,109]

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
