# Peer review of "The Relationship Between Inflammation and the Development of Cerebral Ischaemia and Hypoxia in Traumatic Brain Injury—A Narrative Review"

_ijms, 2025, doi:10.3390/ijms26168066_

Round 1

Reviewer 1 Report

Comments and Suggestions for Authors

The review of the manuscript titled "The relationship between inflammation and the development of cerebral ischaemia and hypoxia in traumatic brain injury – a scoping review" by Nimmo and Younsi, intended for the International Journal of Molecular Sciences.

General Comments:

This scoping review presents a thorough and well-delineated integration of the literature on the role of inflammation in cerebral oedema, hypoxia, and ischemia following traumatic brain injury (TBI). The manuscript is well-written and demonstrates a deep understanding of the pathophysiological mechanisms of secondary injury cascades in TBI. The review is relevant, and makes a strong case for expanding our horizon beyond intracranial pressure (ICP) by considering microvascular dysfunction, glymphatic disruption, and inflammatory responses.

However, the paper would benefit from more focused and concise writing in several sections. Although the discussion is thorough, some parts are repeated several times and contain detailed mechanistic descriptions that could be streamlined or combined. Additionally, greater effort should be made to clearly distinguish speculative mechanisms from well-established findings.

The paper is too long (~28 pages), and there is significant redundancy in discussing oedema types, BBB functioning, and inflammatory processes. I advise authors to merge or abridge duplicative segments (e.g., Sections 7–9), and distinctly distinguish between prominent ideas to render it more understandable.

Occasionally, the review reads more like a textbook chapter than it does a critical scoping review. Background information is needed, but in this case, it is prioritized above the synthesis of new evidence. I suggest to authors that information of general inflammatory biology be kept to a minimum and more is placed on bringing together evidence from TBI-study-specific research.

There is no description of the strategy followed to conduct the scoping review (search plan, inclusion/exclusion criteria, number of included studies). Such omission blunts the scientific transparency of the review.

The review does describe several therapeutic approaches but is not followed by an authoritative synthesis of the most hopeful translational strategies or a roadmap for research directions. Kindly add a special section with salient research gaps, clinical implications, and drug targets of potential interest.

Minor Comments:

Good grammar and style, though there are a few very long sentences that could be split for the readers.

Also, table summarizing oedema subtypes and corresponding therapeutic interventions would be helpful.

Please, use consistent terminology (e.g., “ischaemia” vs. “ischemia”; either UK or US English).

Conclusion:

Recommendation: Major Revision

This is a well-researched review that can potentially be a significant contribution to the literature on TBI pathophysiology. With revisions aimed at improving clarity, conciseness, and methodological transparency, the manuscript will be strengthened significantly.

Author Response

Firstly, we would like to thank you for the time you have take to review our manuscript, and the care you have taken in providing very clear, constructive feedback. We agree with all the points you have raised, and have tried to revise the manuscript to align with your suggestions.

  1. We agree that the manuscript was overly long, primarily because there was over-repetition in a number of areas.
  2. The background information in relation to inflammation has been pared back from 2 sections to 1, with a focus on the aspects that are more relevant to TBI.
  3. In sections 7 to 9, we have also tried to minimize the duplication of information.
  4. We have also tried to make it clearer when an idea may have been more speculative, and when it is extrapolated from clinical or preclinical research.
  5. In terms of the nature of the review, we have decided that it was more appropriate to refer to it as a narrative review rather than a scoping review. A key consideration in doing this was that the starting point for the review was influenced by our own clinical and research experiences. That said, we did adopt a search strategy that was designed to address the key questions asked. Our online search primarily utilized PubMed, with a refining and scaffolding approach taken with the key search terms - e.g. inflammation and oedema; inflammation and cerebral oedema; Inflammation and cerebral oedema and traumatic brain injury etc.
  6. We have added an addition section at the end to address what could represent where some of the main areas for future TBI research may lie, and where there may be more promise in relation to translational research 
  7. We have tried to improve clarity by editing overly long sentences.
  8. On your recommendation, we have added a table that summarizes the main subtypes of cerebral oedema.
  9. We have tried to ensure the consistent use of terminology.

We would like to thank you again for the feedback you have provided.

Reviewer 2 Report

Comments and Suggestions for Authors

A well-structured abstract can help readers capture the essential information of the article. Unfortunately, after reading the abstract twice, I am unable to determine the subject matter of this review article. The introduction is poorly written and difficult to follow. It might be interesting to know how the understanding of inflammation evolved over the past 2000 years; however, I could not understand how the description helps us understand the role of inflammation in TBI or stroke. The main text is fragmented and granular in nature. Authors skipping among different subjects without an in-depth discussion. I am having a difficult time trying to determine what this review is about. There are numerous channels and signal pathways briefly mentioned throughout, without clear connections.  

Author Response

Firstly, we would like to thank you for the time you have take to review our manuscript

With the feedback you have provided, together with that f the other reviewers, we have tried to clarify the message that we were trying to convey in this review.

  1. We agree that the abstract was not clear, and obviously that serves as an entry point to the rest of the review. As such, we have done a significant edit of the abstract, so we hope the message is more clear.
  2. The background information in relation to inflammation has been pared back from 2 sections to 1, with a focus on the aspects that are more relevant to TBI. This is basically how inflammation may lead to the development of oedema, plus how inflammation may influence microvascular function. From that, it is possible to see how both of these aspects may cerebral perfusion and brain tissue oxygenation.
  3. We have added an addition section at the end to address what could represent where some of the main areas for future TBI research may lie, and where there may be more promise in relation to translational research 
  4. We have tried to improve clarity by editing overly long sentences.
  5. We have added a table that summarizes the main subtypes of cerebral oedema.

We would like to thank you again for the feedback you have provided.

Round 2

Reviewer 1 Report

Comments and Suggestions for Authors

The authors address all my concerns. 

Author Response

We would like to thank you once again, not just for the time you have taken to review our manuscript, but also for the clear and constructive feedback that you have provided.

Kindest regards,

Alan and Alex